# Prognostic value of the myocardial salvage index measured by T2-weighted and T1-weighted late gadolinium enhancement magnetic resonance imaging after ST-segment elevation myocardial infarction: A systematic review and meta-regression analysis

Benjamin Kendziora[1¤], Marc Dewey[1,2¤]*

1 Department of Radiology, Charité–Universitätsmedizin Berlin, Berlin, Germany, 2 DZHK (German Centre for Cardiovascular Research), partner site Berlin, Berlin, Germany

¤ Current address: Department of Radiology, Charité–Universitätsmedizin Berlin, Berlin, Germany
* marc.dewey@charite.de

## Abstract

In all patients with ST-segment elevation myocardial infarction, risk stratification should be performed before discharge. The measurement of therapy efficiency with magnetic resonance imaging has been proposed as part of the risk assessment, but it has not been adopted widely. This meta-analysis was conducted to summarize published data on the prognostic value of the proportion of salvaged myocardium inside previously ischemic myocardium (myocardial salvage index) measured by T2-weighted and T1-weighted late gadolinium enhancement magnetic resonance imaging after ST-segment elevation myocardial infarction. Random and mixed effects models were used for analyzing the data of 10 studies with 2,697 patients. The pooled myocardial salvage index, calculated as the proportion of non-necrotic myocardium inside edematous myocardium measured by T2-weighted and T1-weighted late gadolinium enhancement MRI, was 43.0% (95% confidence interval: 37.4, 48.6). The pooled length of follow-up was 12.3 months (95% confidence interval: 7.0, 17.6). The pooled incidence of major cardiac events during follow-up, defined as cardiac death, nonfatal myocardial infarction, or admission for heart failure, was 10.6% (95% confidence interval: 5.7, 15.5). The applied mixed effects model showed an absolute decrease of 1.7% in the incidence of major cardiac events during follow-up (95% confidence interval: 1.6, 1.9) with every 1% of increase in the myocardial salvage index. The heterogeneity between studies was considerable ($\tau$ = 21.3). Analysis of aggregated follow-up data after ST-segment elevation myocardial infarction suggests that the myocardial salvage index measured by T2-weighted and T1-weighted late gadolinium enhancement magnetic resonance imaging provides prognostic information on the risk of major cardiac events, but considerable heterogeneity exists between studies.

**Data Availability Statement:** All relevant data are within the manuscript and its Supporting Information files.

**Funding:** We acknowledge support under the Heisenberg Program of the German Research Foundation and the Open Access Publication Fund of Charité – Universitätsmedizin Berlin and the German Research Foundation. The support will be paid to the corresponding author (MD). The funders had no role in study design, data collection and analysis, decision to publish, or preparation of the manuscript.

**Competing interests:** I have read the journal's policy and the authors of this manuscript have the following competing interests: Outside the submitted work, Prof. Dewey received grants from the German Foundation of Heart Research, GE Healthcare, Bracco, Guerbet, Toshiba Medical Systems, Siemens Medical Solutions, Philips Medical Systems, the German Research Foundation, the European Union's research and innovation funding program FP7, as well as personal fees from the German Research Foundation, Guerbet, Cardiac MR Academy Berlin, Bayer Schering, Toshiba Medical Systems, and Springer. This does not alter our adherence to PLOS ONE policies on sharing data and materials.

# Introduction

In patients with ST-segment elevation myocardial infarction (STEMI), risk assessment should be performed early using information available at the time of presentation [1, 2]. The risk stratification should be recalibrated based on information obtained during hospitalization. Patients having a low risk of complications may be candidates for early discharge. Interventions reducing the risk of major cardiac events (MACE) should be considered in high-risk patients. The left ventricular ejection fraction is one of the strongest predictors of survival and should therefore be measured in all patients in addition to the assessment of clinical markers of high risk, including older age, fast heart rate, hypotension, Kilip class > 1, anterior myocardial infarction, previous myocardial infarction, elevated initial serum creatinine, and history of heart failure or peripheral arterial disease. Moreover, noninvasive testing for ischemia, such as exercise testing or pharmacological stress myocardial perfusion, should be performed before discharge in patients who did not undergo primary percutaneous intervention and may be performed in patients with non-infarct artery disease who have undergone successful primary percutaneous intervention of the infarct artery. Several other strategies, including the measurement of therapy efficiency with magnetic resonance imaging (MRI), have been proposed for risk assessment after STEMI; however, these strategies have not been adopted widely, mainly because of unclear performance characteristics [1, 2].

Therapy efficiency is assessed by MRI through quantification of the salvaged myocardium. Salvage of ischemic myocardium is the main objective of emergency therapy in STEMI, so the amount of salvaged myocardium is a valid marker for therapy efficiency [3]. Myocardial salvage is defined as the difference between the previously ischemic myocardium distal to the infarct artery, the so-called area at risk, and the final necrotic myocardium. To compare the therapy efficiency among infarcts of different sizes, the myocardial salvage index can be calculated as the proportion of non-necrotic myocardium inside the area at risk. For assessing the myocardial salvage index with MRI, T2-weighted MRI and T1-weighted late gadolinium enhancement MRI have most commonly been combined and used based on the assumptions that myocardial edema on T2-weighted MRI allows delineating the ischemic area at risk, and that myocardial necrosis on T1-weighted late gadolinium enhancement MRI can be used to delineate the final necrotic infarct size [4].

This meta-analysis was performed to summarize published data on the prognostic value of the myocardial salvage index measured by T2-weighted and T1-weighted late gadolinium enhancement MRI after STEMI.

# Materials and methods

We reported this meta-analysis according to the PRISMA guidelines [5].

## Eligibility criteria

The following inclusion criteria were applied: a) diagnosis of STEMI in the study patients; b) primary percutaneous intervention as emergency therapy; c) MRI in week 1 after STEMI with reporting of the myocardial salvage index measured by T2-weighted and T1-weighted late gadolinium enhancement MRI or, alternatively, the spatial extent of edematous left ventricular myocardium measured by T2-weighted MRI along with the spatial extent of left ventricular necrotic myocardium measured by T1-weighted late gadolinium enhancement MRI, so that it was possible to calculate the myocardial salvage index; d) usage of a volumetric unit compatible to the percentage of left ventricular myocardium for the measurement of edema and necrosis with MRI; e) reporting of the standard deviation, interquartile range, or confidence interval

(CI) for the myocardial salvage index or the spatial extents of edema and necrosis measured by T2-weighted and T1-weighted late gadolinium enhancement MRI; f) follow-up assessment of MACE defined as cardiac death, nonfatal myocardial infarction, or admission for heart failure; and g) English, French, or German as publication language. We excluded animal studies.

The eligibility criteria were determined by the two reviewers and discussed in the research group on noninvasive cardiovascular imaging at Charité–Universitätsmedizin Berlin.

## Search strategy

We searched in the electronic databases MEDLINE (via PubMed), EMBASE (via Ovid), and ISI Web of Science for references published between the inception of the databases and May 15, 2019. We used a search term that was adjusted according to the standards of the respective databases. The full adjusted search terms can be found in S1 Appendix A. The titles and abstracts of references revealed by the database search were screened, and a full-text review of remaining articles was performed. Additionally, we searched in the bibliographies of finally included studies and reviews revealed by the database search for studies that were missed by the database search.

The search term was determined by the two reviewers and discussed in the research group on noninvasive cardiovascular imaging at Charité–Universitätsmedizin Berlin. The search in the databases, the title and abstract review, and the full-text review were performed by one reviewer (BK); ambiguities were resolved by discussion with the second reviewer (MD).

## Data extraction

A datasheet was predefined, and the following information was extracted from every included study: a) title, first author, publishing journal, and year of publication; b) purpose of the study as mentioned by the study authors; c) study design; d) number of included patients in every patient group; e) myocardial salvage index measured by T2-weighted and T1-weighted late gadolinium enhancement MRI or both the spatial extent of edematous left ventricular myocardium measured by T2-weighted MRI and the spatial extent of left ventricular necrotic myocardium measured by T1-weighted late gadolinium enhancement MRI so that the myocardial salvage index could be calculated; f) incidence of MACE during follow-up; and g) length of follow-up.

The datasheet was created by the two reviewers and discussed in the working group on noninvasive cardiovascular imaging at Charité–Universitätsmedizin Berlin. Data extraction was performed by one reviewer (BK); ambiguities were resolved by discussion with the second reviewer (MD).

## Statistical analysis

The myocardial salvage index was calculated as the proportion of non-necrotic myocardium inside edematous myocardium if the studies did not state the myocardial salvage index measured by T2-weighted and T1-weighted late gadolinium enhancement MRI but provided the spatial extent of edematous left ventricular myocardium measured by T2-weighed MRI and the spatial extent of necrotic left ventricular myocardium measured by T1-weighted late gadolinium enhancement MRI.

Random effects models were used to calculate pooled values for the extracted clinical characteristics of the included study populations (age, gender, prevalence of diabetes, prevalence of hypertension, current smoking, and left ventricular ejection fraction), the time period between STEMI and MRI, the myocardial salvage index measured by T2-weighted and T1-weighted late gadolinium enhancement MRI, the length of follow-up, and the incidence of MACE

during follow-up. Study was included as random effect in all random effects models to account for multiple observations per study. To evaluate the difference in the incidence of MACE between studies with a follow-up length of less than 12 months and studies with a follow-up length equal to or more than 12 months, we added the follow-up length as a categorical variable to the random effects model on the incidence of MACE during follow-up.

Afterwards, we evaluated whether a patient group's mean myocardial salvage index measured by T2-weighted and T1-weighted late gadolinium enhancement MRI correlated with the incidence of MACE in this patient group during follow-up using a mixed effects model. The incidence of MACE during follow-up was used as the dependent variable. The myocardial salvage index was included as fixed effect. To correct for differences in the follow-up period, we included the centered length of follow-up as another fixed effect. Study was included as random effect, again to account for multiple observations per study, which was evaluated by applying Cochran's Q test. On request of the reviewers, we performed a heterogeneity analysis. We suspected that a part of the between-study-heterogeneity could be explained by differences in the average cardiovascular risk of the study populations and the used MRI technique. We therefore included two main cardiovascular risk factors (mean age and prevalence of diabetes) and two main MRI technique parameters (timing of MRI and MRI interpretation) in the mixed effects model and compared the heterogeneity with that of the previous mixed effects model without the inclusion of these factors.

Each patient group's result was weighted by the inverse of the squared estimated standard error of the mean of the myocardial salvage index. Statistical significance was assumed for p-values of 0.05 or smaller. We used R (version 3.6.0, 2019, R Foundation of Statistical Computing) for all calculations. Random and mixed effects models were generated using the metafor R package [6].

Statistics were planned in the research group on noninvasive cardiovascular imaging at Charité–Universitätsmedizin Berlin. Statistical analysis was performed by one reviewer (BK), and the results were discussed in the research group.

### Risk of bias assessment

As we included studies with different study designs, we applied different quality assessment tools. According to a systematic review by Zeng et al. [7], the Cochrane Risk of Bias Tool [8] was used for randomized controlled trials, the Newcastle Ottawa Quality Assessment Scale [9] was used for nonrandomized cohort studies and case control studies, and an 18-item tool by Moga et al. [10] was used for case series studies. To test for the risk of publication bias across studies, Begg and Mazumdar's rank correlation test with continuity correction and Egger's regression test were used in the data on the myocardial salvage index in addition to visually inspecting a funnel plot for obvious asymmetry.

The risk of bias assessment was planned by the two reviewers. Application of the tools was done by one reviewer (BK); ambiguities were resolved by discussion with the second reviewer (MD).

### Results

The search in the electronic databases revealed 1625 references. After removal of duplicates, we screened 1191 references for eligible studies. We excluded 1,019 records at the level of title and abstract. Another 163 references were excluded after the full-text review: 22 because the study patients did not have a STEMI diagnosis, 102 because T2-weighted and late gadolinium enhancement MRI was not done or not sufficiently reported to extract the myocardial salvage index, 38 because follow-up assessment of the incidence of MACE, defined as cardiac death,

nonfatal myocardial infarction, or admission for heart failure, was not performed, and one study because of the language of publication. One eligibly study was found by searching the bibliographies of the included studies and reviews revealed by the database search. Thus, we finally included 10 studies [11–20] with a total of 2,697 patients: two randomized controlled trials [14, 15], six nonrandomized cohort studies [11, 13, 16–19], one case control study [12], and one case series study [20]. Fig 1 shows the PRISMA flow chart [5] summarizing the selection process. The extracted clinical characteristics of the study populations and the used MRI technique is summarized in Table 1.

The random effects models revealed a pooled myocardial salvage index measured by T2-weighted and T1-weighted late gadolinium enhancement MRI of 43.0% (95% confidence interval: 37.4, 48.6), a pooled length of follow-up of 12.3 months (95% CI: 7.0, 17.6), and a pooled incidence of MACE during follow-up of 10.6% (95% CI: 5.7, 15.5). Fig 2 shows the mean myocardial salvage index for all patient groups sorted by the incidence of MACE during follow-up in a forest plot.

The mixed effects model showed a negative correlation between a patient group's mean myocardial salvage index measured by T2-weighted and T1-weighted late gadolinium enhancement MRI and the incidence of MACE in this patient group during follow-up. There was an absolute decrease of 1.7% in the incidence of MACE during follow-up (95% CI: 1.6, 1.9) with every 1% of increase in the myocardial salvage index measured by T2-weighted and T1-weighted late gadolinium enhancement MRI. The heterogeneity between studies was considerable ($\tau$ = 21.3). Model details can be found in Table 2.

The inclusion of two main cardiovascular risk factors (mean age and prevalence of diabetes) and two main MRI technique parameters (timing of MRI and MRI interpretation) in the model for the exploration of heterogeneity reduced the unexplained standard deviation between studies by 65.3% to $\tau$ = 7.4. The details of this adjusted model can be found in S4 Table.

The results of the risk of bias assessment in individual studies are summarized in S2 Table. In one study [12], the quality is reduced by the retrospective design and the uncertainty whether the dropouts were similarly distributed in the within-study groups. In four other studies [11, 15, 16, 19], the quality is reduced by a short follow-up length of 6 months. We did not find evidence of publication bias across studies in the data on the myocardial salvage index measured by T2-weighted and T1-weighted late gadolinium enhancement MRI after STEMI by applying Begg and Mazumdar's rank correlation test (z = -1.67, P = 0.097) and Egger's regression test (t = -1.17, P = 0.255) as well as by visually inspecting the created funnel plot for obvious asymmetry (S1 Fig).

## Discussion

This study was conducted to summarize published data on the prognostic value of the myocardial salvage index measured by T2-weighted and T1-weighted late gadolinium enhancement MRI after STEMI. Meta-regression analysis of aggregated published data shows that a high myocardial salvage index measured by T2-weighted and T1-weighted late gadolinium enhancement MRI in week 1 after STEMI is associated with a low incidence of MACE during follow-up and vice versa with considerable heterogeneity between studies.

To the best of our knowledge, no previous meta-analysis has summarized published data on the prognostic value of the myocardial salvage index measured by T2-weighted and T1-weighted late gadolinium enhancement MRI after STEMI. Two clinical studies, which are included in this meta-analysis, compared the myocardial salvage index measured by T2-weighted and T1-weighted late gadolinium enhancement MRI between patients with and

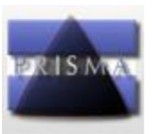

# PRISMA 2009 Flow Diagram

**Identification**

Records identified through database searching
(n = 1625)

Additional records identified through other sources
(n = 1)

Records after duplicates removed
(n = 1192)

**Screening**

Records screened
(n = 1192)

Records excluded
(n = 1019)

**Eligibility**

Full-text articles assessed for eligibility
(n = 173)

Full-text articles excluded
(n = 163):

no diagnosis of STEMI (n = 22)

no use of primary percutaneous intervention as emergency therapy (n = 0)

no MRI in week one after STEMI with reporting of the myocardial salvage index or alternatively the spatial extent of edematous and necrotic left ventricular myocardium measured by T2-weighted and T1-weighted late gadolinium enhancement MRI (n = 82)

no use of a volumetric unit compatible to the percentage of left ventricular myocardium for the measurement of edema and necrosis with MRI (n = 19)

no reporting of the SD, IQR, or CI (n = 1)

no follow-up assessment of MACE defined as cardiac death, nonfatal myocardial infarction, or admission for heart failure (n = 38)

publication not in English, French, or German (n = 1)

**Included**

Studies included
(n = 10)

**Fig 1. PRISMA flow chart.** The search in the electronic databases revealed 1,625 references. A full-text review of 173 studies was performed. Ten studies were found to be eligible and were included in this meta-analysis. STEMI: ST-segment elevation myocardial infarction. MRI: magnetic resonance imaging, SD: standard deviation, IQR: interquartile range, CI: confidence interval, MACE: major cardiac events.

**Table 1. Clinical characteristics of the study populations and the used MRI technique.**

| Clinical characteristics of the study populations | Pooled mean (95% CI) |
|---|---|
| Age, years | 60.5 (95% CI: 58.4, 62.5) |
| Male, % of patients | 80.3 (95% CI: 76.7, 84.0) |
| Diabetes, % of patients | 22.7 (95% CI: 20.9, 24.7) |
| Hypertension, % of patients | 51.7 (95% CI: 42.3, 61.2) |
| Dyslipidemia, % of patients | 32.0 (95% CI: 24.8, 39.3) |
| Current smoking, % of patients | 50.5 (95% CI: 43.5, 57.5) |
| Left ventricular ejection fraction, % | 51.4 (95% CI: 49.3, 53.5) |
| **MRI technique used in the studies** | **Pooled mean (95% CI)** |
| Timing of MRI, days after STEMI | 4.6 (95% CI: 3.2, 6.0) |
| T2-weighted MRI sequence[a] | |
| T2-weighted dark-blood TSE/FSE with IR (STIR) | 10 studies (2,697 patients) |
| T1-weighted late gadolinium enhancement MRI sequence[b] | |
| IR or PSIR using segmented FLASH readout (SPGR) | 9 studies (2,393 patients) |
| IR with single-shot SSFP | 1 study (304 patients) |
| MRI interpretation[c] | |
| Signal intensity > 2 SD above remote myocardium for delineating myocardial edema on T2-weighted MRI and > 5 SD above remote myocardium for quantifying myocardial necrosis on T1-weighted late gadolinium enhancement MRI | 6 studies (1,970 patients) |
| Manual contouring for both delineating myocardial edema on T2-weighted MRI and quantifying myocardial necrosis on T1-weighted late gadolinium enhancement MRI | 4 studies (727 patients) |
| Type of gadolinium contrast agent | |
| Gadobutrol | 4 studies (1,404 patients) |
| Gadopentetate | 3 studies (700 patients) |
| Gadoterate | 2 studies (247 patients) |
| Gadobutrol or gadopentetate | 1 study (346 patients) |
| Dose of gadolinium contrast agent | |
| 0.15 mmol/kg | 5 studies (1,522 patients) |
| 0.2 mmol/kg | 4 studies (871 patients) |

(*Continued*)

**Table 1.** (Continued)

| | |
|---|---|
| 0.1 mmol/kg | 1 study (304 patients) |

MRI: magnetic resonance imaging, STEMI: ST-segment elevation myocardial infarction, MACE: major cardiac events, SD: standard deviation, TSE: turbo spin echo, FSE: fast spin echo, IR: inversion recovery, STIR: short tau inversion recovery, SSFP: steady state free precession, ACUTE: Acquisition for Cardiac Unified T2 Edema, PSIR: phase sensitive inversion recovery, FLASH: fast low angle shot, SPGR: spoiled gradient echo, MD: magnetization driven, FWHM: full width at half maximum, OAT: Otsu's Automated Technique, FACT: automated feature analysis and combined thresholding infarct sizing.

[a]Categories: T2-weighted dark-blood TSE/FSE with IR (STIR), T2-prepared bright-blood single-shot balanced SSFP, hybrid TSE-SSFP (ACUTE), BLADE k-space coverage for dark-blood TSE.

[b]Categories: IR or PSIR using segmented FLASH readout (also referred to as SPGR), IR or PSIR with single-shot SSFP, MD steady state FLASH.

[c]Categories: signal intensity > 2 SD above remote myocardium, signal intensity > 3 SD above remote myocardium, signal intensity > 5 SD above remote myocardium, manual threshold, FWHM algorithm, manual contouring, OAT, FACT algorithm, Heiberg's method.

without occurrence of MACE during follow-up. Eitel et al. and de Waha et al. found a significant difference in the myocardial salvage index between patients with and without MACE during follow-up [12, 14]. De Waha et al. additionally identified the myocardial salvage index as an independent predictor for the incidence of MACE after adjusting for all traditional outcome parameters [12].

As stated in the introduction, the current American College of Cardiology/American Heart Association guideline for STEMI and the current European Society of Cardiology guideline for STEMI recommend risk stratification in all patients hospitalized for STEMI. As a part of the risk assessment, the resting left ventricular ejection fraction should always be measured before discharge, as it is one of the strongest prognostic predictors [1, 2]. Measurement of the resting left ventricular ejection fraction and valve function along with left ventricular thrombus assessment is most commonly performed by echocardiography [21]; however, cardiac cine MRI sequences can also be applied for this purpose [22]. The combination of cine sequences with T2-weighted and T1-weighted late gadolinium enhancement MRI may be used to assess the left ventricular ejection fraction, valve function, existence of a left ventricular thrombus, and the myocardial salvage index as additional prognostic parameter in one examination. Whether additional routine assessment of the myocardial salvage index using cardiac MRI improves the long-term outcome in STEMI patients by more effectively identifying patients who need intensified support or interventions could be studied in a randomized controlled trial.

This meta-analysis has limitations. First, a review protocol was not registered a priori, and so the likelihood that our post hoc decisions are biased is increased [5]. Second, one reviewer conducted the systematic search in the electronic databases, the data extraction, statistical analysis, and risk of bias assessment. This resulted in a higher likelihood of errors in these processes [5, 23]. Third, whether myocardial edema measured by T2-weighted MRI accurately delineates the previously ischemic area at risk is a controversial discussion; thus, whether the myocardial salvage index measured by T2-weighted and T1-weighted late gadolinium enhancement MRI provides an exact measure of the proportion of salvaged myocardium inside the previously ischemic area at risk is unclear [24]. Fourth, we decided against excluding study designs from our analysis, which increases the risk of bias in the analysis; however, we did not find publication bias in the data. Last, the statistical analysis revealed considerable

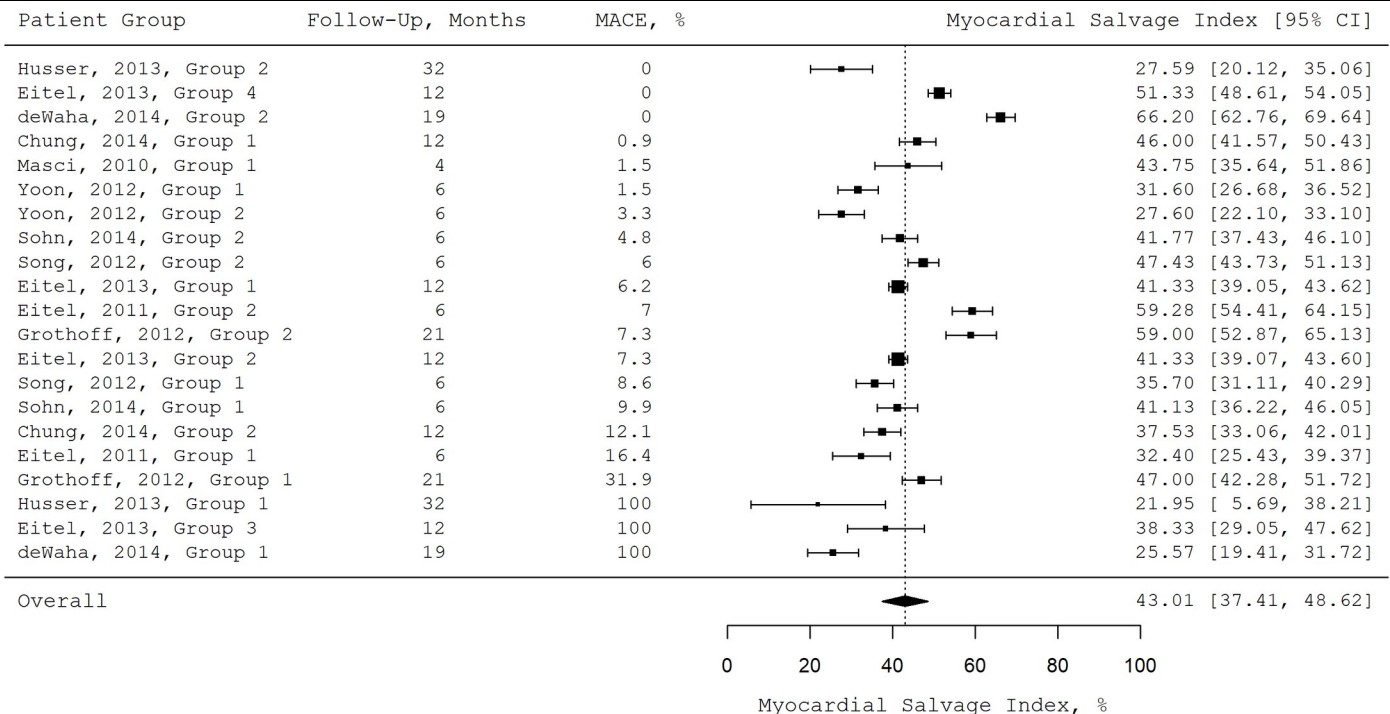

**Fig 2. Forest plot.** Mean myocardial salvage index, length of follow-up, and the incidence of MACE during follow-up for all patient groups. MACE: major cardiac events.

heterogeneity between the included studies. Therefore, exact thresholds of the myocardial salvage index for a low or high risk of MACE cannot be provided.

As a large part of the between-study heterogeneity could be reduced by including cardiovascular risk factors and MRI technique parameters in the meta-regression model, the myocardial salvage index should be interpreted in conjunction with cardiovascular risk factors and the used MRI technique when applied for prognostic purposes. T2- and T1-weighted mapping MRI, as a relatively new and increasingly used alternative to conventional T2-weighted and

**Table 2. Model parameters of the mixed effects model.**

| Dependent variable | | | |
|---|---|---|---|
| Incidence of MACE during follow up, % of patients | | | |
| **Random effects** | | | |
| **Factor** | $\tau^2$ | $\tau$ | **Cochran's Q test for heterogeneity** |
| | | | **p** |
| study | 452.32 | 21.27 | < 0.001 |
| **Fixed effects** | | | |
| **Factor** | **Estimate** | **p** | **Lower 95% CI** | **Upper 95%CI** |
| (Intercept) | 84.56 | < 0.001 | 70.24 | 98.88 |
| Myocardial salvage index, % | -1.73 | < 0.001 | -1.85 | -1.60 |
| Centered length of follow-up, months[a] | 0.42 | 0.593 | -1.13 | 1.98 |

MACE: major cardiac events, CI: confidence interval.

[a]The centered length of follow-up was included into the model to correct for differences in the follow-up length among studies. There was a 9.5% difference in the incidence of MACE during follow-up (95% CI: 1.2, 17.8; P = 0.024) between studies with a follow-up length of less than 12 months and studies with a follow-up length equal to or more than 12 months.

T1-weighted late gadolinium enhancement MRI, allows a more consistent and less subjective delineation of edematous and fibrotic myocardium and may therefore reduce heterogeneity induced by differences in the used MRI technique between study sites in the future [25]. We decided to search for studies that measured myocardial oedema and necrosis with conventional T2-weighted and T1-weighted late gadolinium enhancement MRI, as only a few studies have applied mapping MRI for measuring the myocardial salvage index so far, and we were aiming to include enough data for a valid meta-regression analysis.

In conclusion, analysis of aggregated follow-up data after STEMI suggests that the myocardial salvage index measured by T2-weighted and T1-weighted late gadolinium enhancement MRI provides prognostic information on the risk of MACE, but considerable heterogeneity exists between studies.

## Supporting information

**S1 Appendix A. Search terms.**
(DOCX)

**S1 Table. Raw data.**
(DOCX)

**S2 Table. Risk of bias in individual studies.**
(DOCX)

**S3 Table. PRISMA checklist.**
(DOCX)

**S4 Table. Exploration of heterogeneity.**
(DOCX)

**S1 Fig. Funnel plot.**
(DOCX)

## Author Contributions

**Conceptualization:** Benjamin Kendziora, Marc Dewey.

**Data curation:** Benjamin Kendziora.

**Formal analysis:** Benjamin Kendziora.

**Funding acquisition:** Marc Dewey.

**Methodology:** Benjamin Kendziora.

**Project administration:** Marc Dewey.

**Resources:** Marc Dewey.

**Supervision:** Marc Dewey.

**Visualization:** Benjamin Kendziora.

**Writing – original draft:** Benjamin Kendziora.

**Writing – review & editing:** Marc Dewey.

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
