## [Decision Letter · Decision Letter 0]

30 Oct 2019

PONE-D-19-26741

Prognostic value of the myocardial salvage index measured by T2-weighted and T1-weighted late gadolinium enhancement magnetic resonance imaging after ST-segment elevation myocardial infarction: A systematic review and meta-regression analysis

PLOS ONE

Dear Professor Dewey,

Thank you for submitting your manuscript to PLOS ONE. After careful consideration, we feel that it has merit but does not fully meet PLOS ONE’s publication criteria as it currently stands. Therefore, we invite you to submit a revised version of the manuscript that addresses the points raised during the review process.

In particular, I agree with reviewer 1 that the manuscript would be enhanced by a forest plot summarising the results of the analysis, as is provided in most meta-analyses.

We would appreciate receiving your revised manuscript by Dec 14 2019 11:59PM. To enhance the reproducibility of your results, we recommend that if applicable you deposit your laboratory protocols in protocols.io, where a protocol can be assigned its own identifier (DOI) such that it can be cited independently in the future. For instructions see: http://journals.plos.org/plosone/s/submission-guidelines#loc-laboratory-protocols

We look forward to receiving your revised manuscript.

Kind regards,

Ify Mordi

Academic Editor

PLOS ONE

Journal Requirements:

2. Please ensure that all items in the PRISMA checklist have been performed in your manuscript. For instance, please provide i) In the main text and in the PRISMA diagram, please report the detailed reasons for excluding articles at each stage of your systematic review ; ii) Please discuss in the main text the results of the quality assessment of individual studies. Thank you for your attention to these requests.

I have read the journal's policy and the authors of this manuscript have the following competing interests: Outside the submitted work, Prof. Dewey received grants from German Foundation of Heart Research, grants from GE Healthcare, grants from Bracco, grants from Guerbet, grants from Toshiba Medical Systems, grants from Siemens Medical Solutions, grants from Philips Medical Systems, grants from German Research Foundation (DFG), grants from European Union, FP7, personal fees from German Research Foundation (DFG), personal fees from Guerbet, personal fees from Cardiac MR Academy Berlin, personal fees from Bayer-Schering, personal fees from Toshiba Medical Systems, and personal fees from Springer.

We acknowledge support from the German Research Foundation (DFG) and the Open Access Publication Fund of Charité – Universitätsmedizin Berlin.

In case of publication in Plos One, the fee will be covered by the German Research

Foundation (DFG) and the Open Access Publication Fund of Charité -

Universitätsmedizin Berlin. The support will be paid to the corresponding author (MD).

The funders had no role in study design, data collection and analysis, decision to

publish, or preparation of the manuscript.

Reviewers' comments:

Reviewer's Responses to Questions

**Comments to the Author**

1. Is the manuscript technically sound, and do the data support the conclusions?

Reviewer #1: Yes

Reviewer #2: Yes

2. Has the statistical analysis been performed appropriately and rigorously? 

Reviewer #1: Yes

Reviewer #2: Yes

3. Have the authors made all data underlying the findings in their manuscript fully available?

Reviewer #1: Yes

Reviewer #2: Yes

4. Is the manuscript presented in an intelligible fashion and written in standard English?

Reviewer #1: Yes

Reviewer #2: Yes

5. Review Comments to the Author

Reviewer #1: est map should be provided. As an important element in meta-analysis, the forest map describes the combined effect size and confidence interval for multiple studies.

2. You have mentioned that there was considerable heterogeneity between studies (The last line, page2). Therefore, regress analysis should be used to figure out the reason for heterogeneity.

3. There are a significant number of studies about prognosis after ST-elevation myocardial infarction after 2014, what is the reason that none of these is included in the current study?

Minor weaknesses:

1、 In Page4 line24, you said that your search deadline wass May 15, 2019. What is the search starting date?

2、 In the article you have used different effect models for different parameters. How do you determine when to use a random model or a fixed model？

3、 In order to prove that there is no publication bias in the included article, a funnel chart or other chart proof should be provided.

4、 Limitations should be grouped into a single paragraph.

5、 There are minor linguistic issues which warrant a linguistic revision by a native speaker.

Reviewer #2: The authors analyzed the published data on the prognostic value of the myocardial salvage index measured by T2-weighted and T1-weighted late gadolinium enhancement MRI after STEMI. The main results are clearly presented. However, I have some concerns in this paper listed below.

1) These are studies for examining the prognostic value of the myocardial salvage index, but the patients are highly selected from nonrandomized cohort studies and case control studies. I would urge the authors to present the more detailed information on diagnoses.

2) As the authors indicated, the heterogeneity may partly be explained by differences in MRI sequences. The authors should consider that the studies chosen rely on several different settings for MRI, which all have their own contrast-agent type and dose. Furthermore, the authors are unable to choose with age- and sex-matched studies.

3) The pooled length of follow-up was 12.3 months. The authors need also subgroup analyses between early period and late period (e.g. more than 12 months and less than 12 months).

6. PLOS authors have the option to publish the peer review history of their article (what does this mean?). If published, this will include your full peer review and any attached files.

Reviewer #1: No

Reviewer #2: No

---

## [Author Response · Author response to Decision Letter 0]

17 Dec 2019

Academic Editor

Comment #1: In particular, I agree with reviewer 1 that the manuscript would be enhanced by a forest plot summarising the results of the analysis, as is provided in most meta-analyses.

We agree with you and have added a forest plot in the manuscript, which shows the mean myocardial salvage index measured by T2-weighted and T1-weighted late gadolinium enhancement magnetic resonance imaging (MRI) with confidence interval for every included patient group and the pooled value with confidence interval across all patient groups. To visualize the main result of our meta-regression analysis (negative correlation between a patient group’s mean myocardial salvage index and the incidence of major cardiac events [MACE] in this patient group during follow-up), we added the incidence of MACE during follow-up to the forest plot and sorted the patient groups by the incidence of MACE. Readers can now visually perceive the increasing incidence of MACE during follow-up with decreasing myocardial salvage index, as well as the considerable heterogeneity between studies. We are very thankful for this valuable comment.

Formatting and requirements of the journal.

For the sake of conciseness, we do not repeat here your advice on formatting and how to meet the journal’s requirements, but we have strictly followed your instructions. In particular, we have reformatted the manuscript according to the PLOS ONE style template, detailed the reasons for excluding articles, discussed the quality assessment of individual studies in the main text, added a sentence to the statement on competing interests, removed the funding-related text from the main text, and uploaded our figures to the Preflight Analysis and Conversion Engine. Thank you for the advice on how to meet the criteria for publication.

Reviewer #1

Comment #1: Forest map should be provided. As an important element in meta-analysis, the forest map describes the combined effect size and confidence interval for multiple studies.

Thank you for your suggestion. We have added a forest plot. Please refer to our answer to comment #1 of the Academic Editor for further details.

Comment #2: You have mentioned that there was considerable heterogeneity between studies (The last line, page2). Therefore, regress analysis should be used to figure out the reason for heterogeneity.

We agree with you and have added a heterogeneity analysis. We have included two main cardiovascular risk factors (mean age and prevalence of diabetes) and two main MRI technique parameters (timing of MRI and MRI interpretation) in the meta-regression model, which reduced the unexplained standard deviation between studies τ by 65 %.

Comment #3. There are a significant number of studies about prognosis after ST-elevation myocardial infarction after 2014, what is the reason that none of these is included in the current study?

We were also surprised that we did not find an eligible study published between 2015 and May 15, 2019. We found studies published during this period that performed MRI in week 1 after ST-segment elevation myocardial infarction (STEMI) with reporting of the myocardial salvage index measured by T2-weighted and T1-weighted late gadolinium enhancement MRI or sufficient data to calculate the myocardial salvage index; however, none of these studies assessed and stated the incidence of MACE during follow-up defined as cardiac death, nonfatal myocardial infarction, or admission for heart failure. On the other hand, there are prognostic studies published during this period that did not perform and report the results of T2-weighted and T1-weighted late gadolinium enhancement MRI after STEMI.

Minor weakness #1: In Page4 line24, you said that your search deadline was May 15, 2019. What is the search starting date?

We searched the databases from their date of inception until May 15, 2019. The start of the search is now specified in the text. 

Minor weakness #2: In the article you have used different effect models for different parameters. How do you determine when to use a random model or a fixed model?

If a study provided results for sub-groups, we included these results separately in our analysis. We did not assume these observations to be independent from each other because of similar study methods and therefore included study as random effect into all models. Consequently, we used random effects models instead of fixed effects models to calculate pooled values and a mixed effects model instead of a fixed effects model for the meta-regression analysis. In the manuscript, we did not mention how we chose the models, but we have now added it as a result of this helpful comment.

Minor weakness #3: In order to prove that there is no publication bias in the included article, a funnel chart or other chart proof should be provided.

We agree with you and have added a funnel plot, which we inspected for apparent asymmetry.

Minor weakness #4: Limitations should be grouped into a single paragraph.

Thank you. We have done so accordingly.

Minor weakness #5: There are minor linguistic issues which warrant a linguistic revision by a native speaker.

Thank you for the advice. We have availed of professional proofreading services by an editing company, which helped to improve our paper considerably.

Reviewer #2

Comment #1: These are studies for examining the prognostic value of the myocardial salvage index, but the patients are highly selected from nonrandomized cohort studies and case control studies. I would urge the authors to present the more detailed information on diagnoses.

Thank you for the advice. We extracted details on included patients (age, gender, prevalence of diabetes, hypertension, dyslipidemia, current smoking, and left ventricular ejection fraction) and the used MRI technique (timing of MRI, applied MRI sequences, MRI interpretation, type of contrast agent, and dose of contrast agent) from every included study and summarized these information in a newly created table.

Comment #2: As the authors indicated, the heterogeneity may partly be explained by differences in MRI sequences. The authors should consider that the studies chosen rely on several different settings for MRI, which all have their own contrast-agent type and dose. Furthermore, the authors are unable to choose with age- and sex-matched studies.

We agree that differences in the MRI settings and study populations among the included studies should be considered. The revised manuscript thus includes a heterogeneity analysis. We evaluated whether the inclusion of two main cardiovascular risk factors (age and diabetes) and two main MRI technique parameters (timing of imaging and MRI interpretation) in the meta-regression model reduced the between-study heterogeneity, which it did by 65%. We agree that differences in sex distribution, contrast type, and contrast dose may also explain a part of the heterogeneity. However, we focused on the mentioned parameters and did not include all available parameters in the heterogeneity analysis to avoid overfitting of the meta-regression model.

Comment #3: The pooled length of follow-up was 12.3 months. The authors need also subgroup analyses between early period and late period (e.g. more than 12 months and less than 12 months).

We share your view that the effect of the length of follow-up on the incidence of MACE during follow-up should be taken into account. Therefore, we have included the centered length of follow-up as a fixed effect into the meta-regression analysis to correct for differences in the follow-up length among studies. We have added a subgroup analysis to compare the incidence of MACE in studies with a follow-up length of less than 12 months with the incidence of MACE in studies with a follow-up length of 12 months or longer.

---

## [Decision Letter · Decision Letter 1]

23 Jan 2020

Prognostic value of the myocardial salvage index measured by T2-weighted and T1-weighted late gadolinium enhancement magnetic resonance imaging after ST-segment elevation myocardial infarction: A systematic review and meta-regression analysis

PONE-D-19-26741R1

Dear Dr. Dewey,

We are pleased to inform you that your manuscript has been judged scientifically suitable for publication and will be formally accepted for publication once it complies with all outstanding technical requirements.

With kind regards,

Ify Mordi

Academic Editor

PLOS ONE

Additional Editor Comments (optional):

The authors have adequately addressed all comments suggested from the original submission, and the manuscript is much improved.

Reviewers' comments:

Reviewer's Responses to Questions

**Comments to the Author**

1. If the authors have adequately addressed your comments raised in a previous round of review and you feel that this manuscript is now acceptable for publication, you may indicate that here to bypass the “Comments to the Author” section, enter your conflict of interest statement in the “Confidential to Editor” section, and submit your "Accept" recommendation.

Reviewer #1: All comments have been addressed

Reviewer #2: All comments have been addressed

2. Is the manuscript technically sound, and do the data support the conclusions?

Reviewer #1: Yes

Reviewer #2: Yes

3. Has the statistical analysis been performed appropriately and rigorously? 

Reviewer #1: Yes

Reviewer #2: Yes

4. Have the authors made all data underlying the findings in their manuscript fully available?

Reviewer #1: Yes

Reviewer #2: Yes

5. Is the manuscript presented in an intelligible fashion and written in standard English?

Reviewer #1: Yes

Reviewer #2: Yes

6. Review Comments to the Author

Reviewer #1: All suggestions from both reviewers have been adequately addressed. The quality of the manuscript has been substantial improved comparing to the original version.

Reviewer #2: No further comments. This second version of the paper is a great improvement. I believe the paper will be of interest to the readership of PLOS ONE.

7. PLOS authors have the option to publish the peer review history of their article (what does this mean?). If published, this will include your full peer review and any attached files.

Reviewer #1: Yes: Minjie Lu

Reviewer #2: No

---

## [Editor Report · Acceptance letter]

28 Jan 2020

PONE-D-19-26741R1 

Prognostic value of the myocardial salvage index measured by T2-weighted and T1-weighted late gadolinium enhancement magnetic resonance imaging after ST-segment elevation myocardial infarction: A systematic review and meta-regression analysis 

Dear Dr. Dewey:

I am pleased to inform you that your manuscript has been deemed suitable for publication in PLOS ONE. Congratulations! Your manuscript is now with our production department. 

With kind regards,

on behalf of

Dr. Ify Mordi 

Academic Editor

PLOS ONE